# Underwater Submarine Path Planning Based on Artificial Potential Field Ant Colony Algorithm and Velocity Obstacle Method

**DOI:** 10.3390/s22103652

**Published:** 2022-05-11

**Authors:** Jun Fu, Teng Lv, Bao Li

**Affiliations:** Department of Navigation Engineering, Naval University of Engineering, Wuhan 430000, China; 13871164852@163.com (J.F.); 13545299580@163.com (B.L.)

**Keywords:** underwater, path planning, artificial potential field, ant colony algorithm, velocity obstacle method

## Abstract

Navigating safely in complex marine environments is a challenge for submarines because proper path planning underwater is difficult. This paper decomposes the submarine path planning problem into global path planning and local dynamic obstacle avoidance. Firstly, an artificial potential field ant colony algorithm (APF-ACO) based on an improved artificial potential field algorithm and improved ant colony algorithm is proposed to solve the problem of submarine underwater global path planning. Compared with the Optimized ACO algorithm proposed based on a similar background, the APF-ACO algorithm has a faster convergence speed and better path planning results. Using an inflection point optimization algorithm greatly reduces the number and length of inflection points in the path. Using the Clothoid curve fitting algorithm to optimize the path results, a smoother and more stable path result is obtained. In addition, this paper uses a three-dimensional dynamic obstacle avoidance algorithm based on the velocity obstacle method. The experimental results show that the algorithm can help submarines to identify threatening dynamic obstacles and avoid collisions effectively. Finally, we experimented with the algorithm in the submarine underwater semi-physical simulation system, and the experimental results verified the effectiveness of the algorithm.

## 1. Introduction

Submarines are ships that can operate independently underwater. Since its appearance, it has been widely used in various military operations and has become an important part of modern naval operations.

With the continuous changes in the international situation, a series of studies on underwater navigation of submarines have received extensive attention. Different to surface ships, submarines face more unknowns and threats underwater, all of which pose challenges to the safe underwater navigation of submarines. In order to ensure the safety of navigation, submarines need to find the path that best meets the mission requirements among the many optional routes. Therefore, a suitable path planning algorithm is particularly important.

Due to the complexity and nonlinearity of the underwater 3D environment, the 2D land-based algorithms cannot be directly used to solve the underwater path planning problems.

In response to this issue, many scholars have focused on the development of underwater path planning algorithms. At present, they can be categorized into graph search-based approaches, such as Dijsktra algorithm [1,2,3], $A^{*}$ algorithm [4,5]; sample planning-based approaches, such as PRM algorithm [6,7], $RRT^{*}$ algorithm [8,9]; artificial potential field (APF)-based approaches [10,11]; evolutionary algorithms (EAs)-based approaches, such as distribution estimation algorithm (EDA) [12], particle swarm optimization (PSO) [13,14], genetic algorithm (GA) [15,16], differential evolution algorithm (DE) [17]; heuristic algorithms (HAs)-based approaches, such as ant colony algorithm (ACO) [18,19], simulated annealing algorithm (SA) [20,21].

Based on these algorithms, a large number of scholars have developed path planning algorithms adapted to the three-dimensional environment or the marine environment.

Arinaga et al. applied Dijkstra algorithm to a global path search for AUVs in an underwater environment. The algorithm can avoid a set of obstacles and reach the end but does not consider the impact on the marine environment [1]. Garau et al. implemented a path search with $A^{*}$ algorithm and considered the influence of marine environmental factors [22]. Carreras et al. employed the $RRT^{*}$ algorithm to perform 2D AUV path planning, and the 3D results show that the adaptability of this method in the real complex environment is satisfactory [8]. Jantapremjit et al. not only realize automatic obstacle avoidance by applying the APF algorithm but also introduced the state-dependent Riccati equation method to optimize the optimal high-order sliding mode control, which improved the robustness of the AUV motion [23]. Liu et al. used the PSO algorithm for AUV path planning. Simulation experiments show that the algorithm is simply easy to implement, not sensitive to the population size, and has a faster convergence speed [13]. Ma et al. introduced alarm pheromones in the ACO algorithm (AP-ACO) for path planning of underwater vehicles. The experimental results show that compared with the ordinary ACO algorithm, AP-ACO has a faster convergence speed and stability [24]. Rafael et al. introduced a vortex field in the improved APF algorithm. This method can effectively reduce the collision risk between the UAV and obstacles and reduce the vibration of the path. In addition, the introduction of the artificial potential field algorithm effectively solves the local minimum and oscillation problems of the threshold [25]. 

It is worth mentioning that, with the gradual complexity of application scenarios, the combination of multiple algorithms has become a hot research direction in the field of path planning.

Zhang et al. proposed a branch selection rapid exploration random tree (BS-RRT) algorithm to solve the global path planning problem in the narrow channel environment of UAVs. However, these two algorithms are aimed at aerial unmanned equipment, which is not the same as the background of this study [26]. Yu et al. combined the improved Grey Wolf Optimization algorithm with the $D^{*}$ Light algorithm and proposed a multi-target path planning algorithm for an unmanned cruise ship in an unknown obstacle environment [27]. The algorithm effectively improves the path planning efficiency of unmanned ships and can achieve good results in complex simulation experimental environments. However, the algorithm is limited to two-dimensional path planning, and its adaptability in a three-dimensional environment has not been further studied. A summary of the algorithm is shown in Table 1.

At present, due to the special mission background, there are few studies on the path planning of submarines. More researchers have turned to the path planning algorithms for underwater vehicles such as AUVs. Although the operating environment of AUVs is similar to that of submarines, there are significant differences between the two in terms of size, mission background, and dynamic characteristics. These differences are summarized in Table 2. Therefore, simply using AUV’s path planning algorithm for submarine path planning is inappropriate and has hidden dangers. This paper aims to develop a path planning algorithm with significant advantages for the particularity of submarines.

Considering the applicability and stability of the algorithm, we based on the relatively mature ant colony algorithm, combined the artificial potential field algorithm, and proposed a new artificial potential field ant colony algorithm (APF-ACO). The algorithm is not a simple combination of several algorithms, but a special development for the characteristics of submarine underwater navigation. The algorithm can effectively deal with the underwater path planning problem of submarines. In this sense, the main contributions of this paper include:In this paper, the APF-ACO algorithm is proposed to solve the underwater global path planning problem of submarines. The algorithm is able to converge rapidly in the underwater environment. Moreover, the path planning results obtained by this algorithm are more advantageous and more stable.In order to further optimize the path planning results obtained by the APF-ACO algorithm and make it more in line with the navigation requirements of submarines, this paper develops an inflection point optimization algorithm and a path smoothing algorithm. Experimental results show that the algorithm significantly reduces the path length and the number of inflection points.A dynamic obstacle avoidance algorithm based on the velocity obstacle method is proposed, which further improves the content of the submarine path planning algorithm. The experimental results show that the dynamic obstacle avoidance algorithm can accurately identify and avoid threatening dynamic obstacles.Discuss the real performance of the algorithm in the submarine semi-physical simulation system. From the results, the real feasibility of the algorithm under the underwater navigation of the submarine is verified.

## 2. Improved Artificial Potential Field Ant Colony Algorithm (APF-ACO)

In this section, we discuss the physical background and derivation of the algorithm in detail.

### 2.1. Models

The environment model simulates the real application environment with an abstract physical and mathematical model. The representation of the model directly affects the efficiency and reliability of the path planning algorithm.

In this paper, the path planning environment is constructed by the method of spatial equal mesh. As shown in Figure 1, the minimum grid corresponds to the space with the minimum longitude, latitude, and depth interval. A plane with a depth value of 0 is taken as an isosurface with a *z*-axis value of 0. Vertical down along the depth is the positive direction of the *z*-axis. The direction of increase along latitude is the positive *x*-axis. The increasing direction along the longitude is the positive y direction.

In this way, we constructed a cuboid underwater planning space. Let’s say we divide space into (l+1) parallel planes along the *x*-axis. Each plane is then divided into (m+1)×(n+1) grids along the *y*- and *z*-axes. In this way, the planning space is divided into (l+1)×(m+1)×(n+1) subspaces.

In the planning space, each point in the subspace can be spatially located with coordinates (x, y, z), as shown in Figure 2. This completes the environmental modeling of the planned space.

### 2.2. Improved Artificial Potential Field Function

#### 2.2.1. Improved Repulsion Function

The original artificial potential field function is prone to deadlock when there are a lot of obstacles around the target point, which makes the algorithm unable to effectively converge. To solve this problem, the repulsive function of the artificial potential field algorithm is improved in this paper. We introduce the distance between node and target point to adjust the size of the repulsive force field to prevent the planning from falling into local deadlock. The improved repulsion function (1) is shown.
(1)$$\{\begin{array}{c} U_{rep}\left(X\right)=\frac{1}{2}k_{rep}{\left(\frac{1}{\rho \left(X,X_{0}\right)}-\frac{1}{\rho_{0}}\right)}^{2}\cdot \xi \;\rho \left(X,X_{0}\right)\leq \rho_{0}\\ 0\;else \end{array}$$
where $\xi ={\left(X-X_{d}\right)}^{n}$ represents the Euclidean distance between the current node and the target point to the n power. $\rho_{0}$ is the influence range value of the obstacle. If the distance between the carrier and the obstacle is greater than $\rho_{0}$, the repulsive force is 0. $\rho \left(X,X_{0}\right)$ is the Euclidean distance between the machine and the obstacle at a certain time. n is a positive adjustment constant. The introduction of ξ can reduce the repulsion effect near the target point so that the carrier can reach the target point smoothly.

By calculating the negative gradient of $U_{rep}\left(X\right)$, the repulsion calculation formula of the improved algorithm can be obtained as:(2)$$F_{rep}\left(X\right)=F_{rep1}\left(X\right)+F_{rep2}\left(X\right)$$
where
(3)$$\{\begin{array}{c} F_{rep1}\left(X\right)=k_{rep}\left(\frac{1}{\rho \left(X,X_{0}\right)}-\frac{1}{\rho_{0}}\right)\cdot \frac{\xi }{\rho^{2}\left(X,X_{0}\right)}\\ F_{rep2}\left(X\right)=-\frac{n}{2}\cdot k_{rep}\cdot {\left(\frac{1}{\rho \left(X,X_{0}\right)}-\frac{1}{\rho_{0}}\right)}^{2}\cdot {\left(X-X_{d}\right)}^{n-1} \end{array}$$

$F_{rep1}\left(X\right)$ and $F_{rep2}\left(X\right)$ describe respectively the repulsion force of the robot pointing at the obstacle and the attraction force of the robot pointing at the target point. Thus, the repulsive function expression of the improved algorithm is constructed.

#### 2.2.2. Improved Repulsion Function

The expression for gravitational potential energy is:(4)$$U_{att}\left(X\right)=\frac{1}{2}k_{p}{\left(X-X_{d}\right)}^{2}=\frac{1}{2}k_{p}\left({\left(x-x_{d}\right)}^{2}+{\left(y-y_{d}\right)}^{2}+{\left(z-z_{d}\right)}^{2}\right)$$
where $k_{p}$ is the target gain coefficient, and the attraction of the gravitational potential field to the carrier is the negative gradient direction of the gravitational potential energy:(5)$$F_{att}=-k_{p}n_{1}\varepsilon_{1}$$
where $n_{1}$ is a unit vector with direction between the vector and the target point. $\varepsilon_{1}=\|X-X_{d}\|$ is the Euclidean distance between the carrier and the target point.

In this way, the resultant force of the artificial potential field can be obtained:(6)$$F_{t}=F_{rep}+F_{att}$$

### 2.3. Artificial Potential Field Ant Colony Optimization (APF-ACO) Algorithm

#### 2.3.1. Heuristic Function

The heuristic function of the traditional ant colony algorithm only considers the distance between the current position and the next node. When there are a large number of obstacles near the endpoint, the algorithm tends to fall into a local optimum. Not only that, but the positive feedback of the ant colony algorithm may also make the final planned path, and not the global optimal path.

Therefore, this paper introduces the resultant force $F_{t}\;$ of the artificial potential field into the design of the heuristic function, so as to achieve the purpose of obtaining the global optimal path. The improved heuristic function is shown in (7).
(7)$$\eta_{ij}\left(t\right)=\eta_{d}\cdot \eta_{F}=\frac{d_{Sj}}{\left(d_{ij}+d_{jG}\right)}\cdot a^{F_{t}\cdot \zeta \cdot \cos\theta }$$
where $\zeta =1-\frac{N_{n}}{N_{max}}$, $N_{n}$ is the number of previous iterations, and $N_{max}$ is the maximum number of iterations.

$\eta_{d}$ is the distance heuristic function. where $d_{Sj}$ is the distance between the starting point S and the next node j. $d_{jG}$ is the distance between the next node j and the target point G. According to this formula, it can be seen that the size of the heuristic function is related to the distance between the starting point and the target point. In this way, the more likely the node j is to be selected. The improved distance heuristic function ensures that the ants always move away from the starting point and close to the target point to prevent the algorithm from falling into deadlock.

$\eta_{F}=a^{F_{t}\cdot \cos\theta }$ is the potential field heuristic function, where $F_{t}$ is the resultant force of the potential field, θ is the angle between the direction of the connection between the current node and the next node and the direction of the potential field force, and a is a positive integer greater than zero. The introduction of artificial potential field parameters can speed up the convergence speed of the ant colony algorithm, but at the same time, it may cause the result to fall into a local optimum in the latter part of the iteration. Therefore, a decay coefficient ζ is introduced to make the potential field force decrease with the continuous iteration of the algorithm.

#### 2.3.2. Pheromone Diffusion Model

The basic ant colony system local pheromone update rule is
(8)$$\tau_{ij}^{\prime }=\left(1-\rho \right)\tau_{ij}^{\prime }+\rho \Delta \tau_{ij}^{\prime }$$

This formula adopts the uniform update rule and does not consider the influence of pheromone diffusion on pheromone distribution, which deviates from the real ant colony system. In order to restore the efficiency of the ant colony system and improve the convergence speed of the original ant colony system algorithm, this paper introduces the pheromone diffusion model to improve the utilization efficiency of pheromone by the ant colony.

We assume that the pheromone concentration follows a Gaussian distribution and that the pheromone at the current point only spreads to the adjacent forward direction grid. The simplified pheromone diffusion model is a circumscribed sphere, as shown in Figure 3.

Where $l_{ob}$ represents the diffusion radius of the pheromone. We assume that the step size of each movement of the ant is 1 in the y and z directions, then $l_{ob}$ is $\sqrt{3}$. In (7), the angle θ of the potential field force is calculated for the adjacent grids of the current grid. The adjacent grid with the smallest θ value is the pheromone diffusion direction. From this, the direction of the next grid j is determined, and the pheromone concentration diffused to point j is calculated as:(9)$$\tau_{ij}^{\prime \prime }=\delta \cdot q\left(i\right)\cdot \frac{l_{ob}-d}{l_{ob}}\;(0<\delta \leq 1)$$
where δ is the diffusion coefficient of the pheromone, and d is the Euclidean distance between the current node i and the next candidate node j.

#### 2.3.3. State Transition Rules

According to the above-improved algorithm, the state transition rule of ants from point i to the next grid point is calculated:(10)$$P_{ij}^{k}\left(t\right)=\frac{\tau_{ij}{}^{\alpha }\left(t\right)\cdot \eta_{ij}^{\beta }\left(t\right)}{\sum_{s\in allowed_{k}}\tau_{ij}{}^{\alpha }\left(t\right)\cdot \eta_{ij}^{\beta }\left(t\right)}$$
where $\tau_{ij}=\tau_{ij}^{\prime }+\tau_{ij}^{\prime \prime }$, $\eta_{ij}$ is obtained by (7).

Through the above steps, the algorithm not only increases the probability of ants choosing j point but also maintains the diversity of paths. Moreover, the convergence speed of the algorithm is accelerated while avoiding the path from falling into the local optimum.

After one iteration is completed, the global pheromone update is performed, and the update rule is as follows.
(11)$$\tau_{ij}^{\prime }\left(t,t+1\right)=\left(1-\rho \right)\tau_{ij}^{\prime }\left(t\right)+\Delta \tau_{ij}^{\prime }\left(t,t+1\right)$$
where
(12)$$\Delta \tau_{ij}^{\prime }\left(t,t+1\right)=\{\begin{array}{c} \frac{1}{L_{gb}},\;\left(i,j\right)\;\in global_{best}\\ 0\;\mathrm{else} \end{array}$$

The specific flow chart of the APF-ACO algorithm proposed in this paper is shown in Figure 4.

## 3. Path Optimization Algorithm

The results obtained by the path planning algorithm usually have shortcomings such as many inflection points and insufficient smoothness. To solve these problems, the path results need to be optimized. In this paper, the inflection points optimization algorithm and the path smoothing algorithm are used to obtain the path results that are more suitable for submarine navigation.

In addition to global static path planning, submarines also need to have dynamic obstacle avoidance capabilities to deal with sudden threats. This paper proposes a three-dimensional dynamic obstacle avoidance algorithm based on the velocity obstacle method to improve the survivability of submarines.

### 3.1. Path Inflection Point Optimization

First, find the inflection points in the path that the algorithm gets. Suppose there are m path points in the grid space along the path planning direction. The current path point is i(1<i<m), and d(i,i−1) is the distance between point i and point i−1. d(i,i+1) is the distance between point i and i+1. d(i−1,i+1) is the distance between point i−1 and point i+1. Then the inflection point can be judged according to the triangle rule:(13)$$\{\begin{array}{c} d\left(i,i+1\right)+d\left(i,i—1\right)=d\left(i-1,i+1\right),\;i\;\notin Inf\\ d\left(i,i+1\right)+d\left(i,i—1\right)>d\left(i-1,i+1\right),\;i\in Inf \end{array}$$

All inflection points of the path can be obtained in this way. Add the start and endpoints, assuming there are n inflection points in total, and arrange all the inflection points in sequence: $Inf_{1},\;Inf_{2},\;Inf_{3}\cdots Inf_{n}$. Connect with the second inflection point from the starting point. If the connected line does not pass-through obstacles, continue to connect with the third inflection point, and so on to connect with the 4th, 5th, nth inflection points. If the starting point and the  kth point (1<k≤n) pass through an obstacle when connecting, connect the starting point with the (k−1)th point as the first segment of the path after optimization. Then connect with the (k−1)th point as the starting point until the target point is reached. The flow chart of the inflection point optimization algorithm is shown in Figure 5. The schematic diagram of the method is shown in Figure 6.

### 3.2. Path Smoothness Optimization

Submarines should try to avoid large steering angles when sailing underwater. In order to meet the actual navigation requirements of submarines, this paper proposes a path smoothing algorithm adapted to the APF-ACO algorithm. The simple single-stage polynomial optimization cannot adapt to the complex underwater environment, and the complex high-order polynomial optimization is not efficient, so this paper chooses to use the Clothoid curve fitting algorithm to optimize the smoothness of the path. The Clothoid curve is based on the Fresnel integral, and the change in curvature of the curve is proportional to the arc length of the curve. The three-dimensional Clothoid curve equations are shown in (14) and (15). In this paper, two-dimensional Clothoid curve fitting is performed on XOY and XOZ in turn, and then the fitting results are combined into three-dimensional fitting results.
(14)$$\{\begin{array}{c} x=x_{0}+h\int_{0}^{s}\cos\left(\psi \left(\tau \right)\right)d\tau \\ y=y_{0}+h\int_{0}^{s}\sin\left(\psi \left(\tau \right)\right)d\tau \end{array}$$
(15)$$\{\begin{array}{c} x=x_{0}+h\int_{0}^{s}\cos\left(\psi \left(\tau \right)\right)d\tau \\ z=z_{0}+h\int_{0}^{s}\sin\left(\psi \left(\tau \right)\right)d\tau \end{array}$$
where ($x_{0}$, $y_{0}$, $z_{0}$) is the starting point coordinate, ψ(τ) represents the tangent angle of the curve, and the expression is:(16)$$\psi \left(\tau \right)=\theta_{0}+k_{0}\tau +\frac{1}{2}c\tau^{2}$$
$\theta_{0}$ is the initial tangent angle, $k_{0}$ is the initial curvature, and s is the arc length of the curve.

Suppose the path point obtained by the APF-ACO algorithm is $P\left(x_{i},y_{i}\right)$ (i=1, 2, 3⋯k). Using the Clothoid curve to fit is to solve the Clothoid curve segment between the k path points under the condition of continuous curvature. Taking the first segment of the path as an example, the coordinates of the endpoints at both ends are $\left(x_{l},y_{l}\right)$, $\left(x_{l+1},y_{l+1}\right)$. According to (14), the two ends should meet the following conditions:(17)$$\{\begin{array}{c} x_{l+1}=x_{l}+h\int_{0}^{s_{l}}\cos\left(\psi_{l}\left(\tau \right)\right)d\tau \\ y_{l+1}=y_{l}+h\int_{0}^{s_{l}}\sin\left(\psi_{l}\left(\tau \right)\right)d\tau \end{array}$$
(18)$$\{\begin{array}{c} x_{l+1}=x_{l}+h\int_{0}^{s_{l}}\cos\left(\psi_{l}\left(\tau \right)\right)d\tau \\ z_{l+1}=z_{l}+h\int_{0}^{s_{l}}\sin\left(\psi_{l}\left(\tau \right)\right)d\tau \end{array}$$
where $\psi_{l}\left(\tau \right)=\theta_{0l}+k_{0l}\tau +\frac{1}{2}c_{l}\tau^{2}$. $s_{l}$ is the arc length of the lth path. To ensure the continuity of curvature at the inflection point, $\left(x_{l+1},y_{l+1}\right)$ should satisfy:(19)$$\{\begin{array}{c} k_{0l+1}=k_{0l}+c_{l}s\\ \theta_{0l+1}=\theta_{0l}+k_{0l}s_{l}+\frac{1}{2}c_{l}s_{l}{}^{2} \end{array}$$

This formula determines that the tangent angle θ and the curvature k of the paths of the lth and l+1th sections at the inflection point are equal.

After initially completing the path smoothness optimization, it is also necessary to consider that the optimized path may collide with obstacles again. Therefore, the smooth path must be re-examined. As shown in Figure 7, the initial smooth path is detected. If the path collides with an obstacle, the curvature k of the fitted path is reduced for re-planning until the smooth path does not collide with the obstacle.

The flowchart of the smoothing optimization algorithm is shown in Figure 8.

### 3.3. Local Dynamic Obstacle Avoidance by Speed Obstacle Method

The velocity obstacle method is an algorithm that uses geometric constraints to express methods to avoid collisions with obstacles. Its schematic diagram is shown in Figure 9. Where $P_{A}$ is the carrier position, $P_{B}$ is the obstacle position, $V_{A}$ is the carrier velocity, $V_{B}$ is the obstacle velocity, $V_{AB}=V_{A}-V_{B}$ is the relative velocity of $V_{A}$ and $V_{B}$. The basic principle of the algorithm is to construct the velocity obstacle area by obtaining the position and velocity information of the carrier and the obstacle. Then, it is determined whether the carrier will collide with the obstacle by calculating whether the relative speed of the carrier and the obstacle is within the space of the obstacle area.

We inflate the submarine into a sphere of radius $r_{A}$ and the obstacle into a sphere of radius $r_{B}$, then the radius of the obstacle sphere is $R_{O}=r_{A}+r_{B}$. It is defined that multiple rays drawn from the geometric center of the carrier are tangent to the expanded spherical obstacle area, and all the tangents form a triangular pyramid space. The β is the angle between $V_{AB}$ and the axis $P_{A}P_{B}$ and the α is 1/2 of the size of the cone apex angle. β and α are shown as (20) and (21).
(20)$$\beta =\arccos\left(\frac{V_{AB}\cdot P_{A}P_{B}}{\|V_{AB}\|\cdot d}\right)$$
(21)$$\alpha =\arcsin\left(\frac{R_{O}}{d}\right)$$
where d is the distance between $P_{A}$ and $P_{B}$.

When β<α, there is a risk of collision, and collision avoidance measures should be taken; when β≥α, there is no risk of collision. Through this method, the dynamic obstacle avoidance problem of the carrier can be simplified to the static obstacle avoidance problem.

We assume that the velocity vector of the carrier and the obstacle do not change during the calculation process, then define a ray from the center of the carrier along the relative velocity direction:(22)$$R\left(P_{A},\;V_{AB}\right)=\left\{P_{A}+V_{AB}\cdot t|t\geq 0\right\}$$
where t represents time and $R\left(P_{A},\;V_{AB}\right)$ represents the ray composed of the current position of the carrier and the direction of the relative velocity vector. Then the collision conditions between the carrier and the obstacle are:(23)$$R\left(P_{A},\;V_{AB}\right)\cap^{\;}P_{O}\neq \emptyset$$

The relative velocities with collision risk are grouped together, which constitutes the “collision domain” of dynamic obstacle avoidance. The mathematical description of the “collision domain” is:(24)$$Z_{c}=\left\{V_{AB}|R\left(P_{A},\;V_{AB}\right)\cap^{\;}P_{O}\neq \emptyset \right\}$$

The velocity obstacle method uses a definite value when describing the dynamic obstacle, but there is a certain inevitable error between the sensor accuracy and the submarine’s own speed, so the submarine may accidentally collide with the obstacle. In order to solve this hidden danger, this paper decided to introduce the parameter of “safe distance”. Compared with the size of the submarine itself and the size of the obstacles, the marine navigation environment is very broad, so the parameter of “safety distance” has practical application value and feasibility. The “safety distance” radius is given by empirical values of sensor accuracy error and submarine speed error.

The corrected obstacle area radius is:(25)$$R_{O}=r_{A}+r_{B}+r_{s}$$

In addition, in order to enhance the adaptability of the algorithm, this paper supplements the dynamic obstacle avoidance algorithm. After dynamic obstacle avoidance, if the current node cannot safely reach the next node of the original path planning result, the current node is used as the starting point for re-path planning. Repeat this operation until the submarine reaches the end. The flow chart of the dynamic obstacle avoidance algorithm is shown in Figure 10.

## 4. Test and Analysis

In this paper, a 10 × 10 × 10 (excluding boundary) three-dimensional space environment is constructed for simulation experiments, in which spheres represent obstacles. In order to verify the effectiveness of the APF-ACO algorithm proposed in this paper, relevant experiments are designed in this section.

### 4.1. Algorithm Performance Comparison Experiment

In order to verify the effectiveness of the APF-ACO algorithm, the experiments designed in this paper are compared with the other three algorithms in various obstacle environments. Among them, the Optimized ACO algorithm is an underwater ant colony optimization algorithm proposed by the reference [24].

In this paper, experiments were designed in five obstacle environments, and each experiment was carried out independently 10 times, and statistical significance tests were performed to ensure the reliability of the results. The parameter settings of the algorithm are shown in Table 3.

The experimental results are shown in Figure 11 and Table 4. It can be seen from the experimental results that APF-ACO and Optimized ACO have significant advantages compared with the original ACS and the original APF algorithm. This proves the superiority of these two algorithms. In addition, compared with the Optimized ACO algorithm, APF-ACO has advantages in the performance of the best results and the average results and is also better in the control of the worst results and standard deviations. In summary, the path planning results of the APF-ACO algorithm have better performance and are more stable.

In order to verify the effectiveness of the APF-ACO algorithm proposed in this paper, we further compare the computational time cost of four different algorithms under the same conditions, and the calculation results are shown in Table 5. The experimental equipment parameters in this paper are Core i9 CPU and 16 G running memory.

It can be seen from the experimental results that the calculation time of The Original APF algorithm is the shortest, followed by the calculation time of The Original ACS. The computation time of the APF-ACO algorithm is almost the same as that of the Optimized ACO algorithm, which is about 10% longer than that of The Original ACS. The results show that the two optimization algorithms sacrifice about 10% of the computing time to obtain better path planning results, which have practical application value.

From the above experimental results, it can be seen that the APF-ACO algorithm proposed in this paper and the Optimized ACO algorithm have significant advantages. In order to further compare the performance of the two algorithms, this paper designs experiments to compare the number of path inflection points and the convergence speed of the operation. In five different experimental environments, the visual planning results of the two algorithms are shown in Figure 12.

The establishment of five experimental environments mainly considers the influence of different obstacle distribution patterns on the calculation results of the path planning algorithm. The distribution of obstacles in test environments 1 and 2 is more discrete, the distribution of obstacles in test environment 3 is more concentrated, the distribution density of obstacles near the endpoint is increased in test environment 4, and the distribution density of obstacles near the start point is increased in test environment 5. It can be seen from the path planning results that in the above five test environments, both algorithms can obtain complete and feasible paths. However, it can also be clearly seen that the path obtained by the Optimized ACO algorithm is more tortuous when the obstacles are dense. In order to further compare the pros and cons of the two algorithms, we recorded the average number of inflection points of the planning results of the two algorithms in each experimental environment, and the results are shown in Table 6. 

It can be seen from the results that the path obtained by APF-ACO is straighter and the number of inflection points is reduced by about 18.8%. This is mainly because the APF-ACO algorithm introduces the potential field force parameter so that it can consider the distribution information of obstacles in the operation of each step. This allows the algorithm to avoid obstacles as early as possible. This feature of APF-ACO also fits the real needs of submarines sailing underwater.

Convergence speed is an important indicator to measure the performance of an algorithm. This paper next compares the convergence speed of APF-ACO and Optimized ACO. We conducted 10 independent experiments in experimental environment 3 and averaged the convergence rate results, and the obtained results are shown in Figure 13.

As can be seen from the figure, the convergence speed of APF-ACO is about 47.7% faster than that of Optimized ACO. At the beginning of the iteration, APF-ACO can obtain better initial path results, mainly because the addition of the potential field force parameter can increase the cost difference between the path points to be selected, thereby greatly increasing the probability of selecting a better path. In the later stage of iteration, the convergence result of APF-ACO is significantly better than that of Optimized ACO, mainly because the introduction of the potential field force parameter can always guide the path to find the direction close to the endpoint, avoiding the convergence result falling into the local optimum.

In summary, it can be seen that the APF-ACO algorithm has certain advantages in terms of convergence results and convergence speed and can adapt to the complex underwater path planning application scenarios of submarines.

### 4.2. Path Smoothness Optimization

The simulation experiment of the path smoothing optimization algorithm is carried out in the experimental environment 5. The inflection points of the path are optimized before smooth optimization, and only the necessary inflection points of the path are retained. The final result is shown in Figure 14:

The green dotted line in Figure 14 is the inflection point optimization result. It can be seen intuitively that the inflection point optimization algorithm reduces the number of inflection points and shortens the path length on the premise of ensuring the safety of the path. In order to quantitatively analyze the effect of the inflection point optimization algorithm, we analyzed its performance under five different obstacle environments, and the results are shown in Figure 15. The red implementation in Figure 15 represents the number of inflection points of the original path, and the red dotted line represents the number of optimized inflection points. The inflection point optimization algorithm can effectively shorten the path length by about 21.7% and reduce the number of inflection points by about 53.5%. This verifies the effectiveness of the algorithm.

The purple solid line in Figure 15 is the result of smooth optimization at the inflection point optimization path. It can be seen that the optimized path has no abrupt turning points, and the path is smoother. Not only that, but the introduction of the obstacle avoidance correction algorithm also ensures that the smoothed path will not collide with adjacent obstacles.

### 4.3. Local Dynamic Obstacle Avoidance Experiment

In order to verify the dynamic obstacle avoidance effect of the speed obstacle method, this paper constructs a dynamic obstacle environment for experiments. The experiment introduces dynamic obstacles on the basis of global path planning to verify the effectiveness of the dynamic obstacle avoidance algorithm and its compatibility with the APF-ACO algorithm. The visualization results are shown in Figure 16.

We construct three dynamic obstacle environments to verify the reliability of the dynamic obstacle avoidance algorithm. We show two different perspectives of each path planning result. As shown in Figure 16, the dynamic obstacle moves in the direction of the arrow. The red track indicates that the dynamic obstacle will collide with the original path, and the green track indicates that the dynamic obstacle will not collide with the original path track. It can be seen from the result in the figure that under the effect of the dynamic obstacle avoidance algorithm, the submarine accurately identified the threatening dynamic obstacles and re-planned them to avoid collisions. Dynamic obstacles without threats will not affect the original path trajectory. In experimental environment 2 and experimental environment 3, the re-planning node cannot safely reach the next node of the original planning result under the current step size requirement, so re-planning is carried out. It can be seen that the re-planned path results can overlap with the original path results as much as possible under the premise of successfully avoiding dynamic obstacles.

It can be seen from the experimental results that the dynamic obstacle avoidance algorithm proposed in this paper can achieve the expected effect and has a practical application value.

### 4.4. System Semi-Physical Experiment

In order to further verify the application of the algorithm proposed in this paper in the real environment, we applied APF-ACO to a semi-physical simulation software specially designed for the study of submarine underwater path planning for experiments. The system uses the 0.25° submarine topography data provided by GEBCO (General Bathymetric Chart of the Oceans) to generate virtual submarine topography to simulate the real underwater navigation environment of submarines. In addition, the system adopts the standard SUBOFF full-body submarine model and fully considers various maneuvering rules that submarines have underwater, such as maximum pitch angle, maximum diving depth, limit turning radius, etc.

The path planning range selected for this system test is 14.3563° N to 16.6942° N, and 113.3719° E to 116.1954° E. The starting point coordinates are 16.5079° N, 113.5875° E, and 100 m deep. The ending coordinates are 14.6072° N, 116.0237° E, and 100 m deep. The specific parameters of the APF-ACO algorithm used in this experiment are shown in Table 7.

The specific experimental results are shown in Figure 17.

It can be seen from the experimental results that the submarine can successfully avoid terrain obstacles and reach its destination. The result of the path is relatively straight as a whole, and it can actively maintain the navigation at the same depth, which is in line with the underwater maneuvering characteristics of submarines. The results verify the effectiveness of the APF-ACO algorithm as a submarine underwater path planning algorithm in the real marine environment.

## 5. Conclusions

In this paper, a path planning algorithm suitable for submarine underwater navigation is proposed. This algorithm is a composite method that includes global path generation and local path adjustment. In the global path planning, we presented an improved Artificial Potential Field Ant Colony Optimization (APF-ACO) algorithm to adapt to the underwater path planning needs of submarines. The experimental results showed that APF-ACO can stably obtain path planning results in a variety of experimental environments. Compared with the Optimized ACO, APF-ACO can obtain path results with shorter length, fewer inflection points, and better stability. Not only that, but APF-ACO also has a faster convergence speed, which meets the tactical needs of submarines.

We also propose an inflection point optimization algorithm and a smooth optimization algorithm to further improve the path obtained by APF-ACO, making it more in line with the actual underwater navigation of submarines. The experimental results show that the inflection point optimization algorithm can effectively reduce the unnecessary inflection points of the path and greatly reduce the length of the path. The smoothing optimization algorithm can increase the smoothness of the path without intersecting with obstacles.

In order to make the algorithm more complete, we have added a dynamic obstacle avoidance algorithm based on the motion obstacle method. The introduction of this algorithm can help submarines identify threatening moving obstacles and re-plan local paths. The experimental results show that the dynamic obstacle avoidance algorithm can successfully help submarines to identify and avoid dangerous dynamic obstacles.

We use a professional semi-physical simulation system for scene verification of the APF-ACO algorithm. In the system, we use real seabed topography data and real starting and ending position data to conduct experiments. The experimental results show that the APF-ACO algorithm can be effectively applied in the navigation tasks of submarines and has practical application value.

We insist on affirming the application value of path planning in submarine navigation in the new era. In future research, we will further optimize the submarine’s path planning algorithm. We consider introducing the influence of marine environmental elements on submarine navigation into the path planning algorithm to expand the scope of application of the algorithm. Next, we seek to apply the algorithm to real submarine navigation tasks and collect data in real experiments to further improve the algorithm.

## Figures and Tables

**Figure 1 sensors-22-03652-f001:**
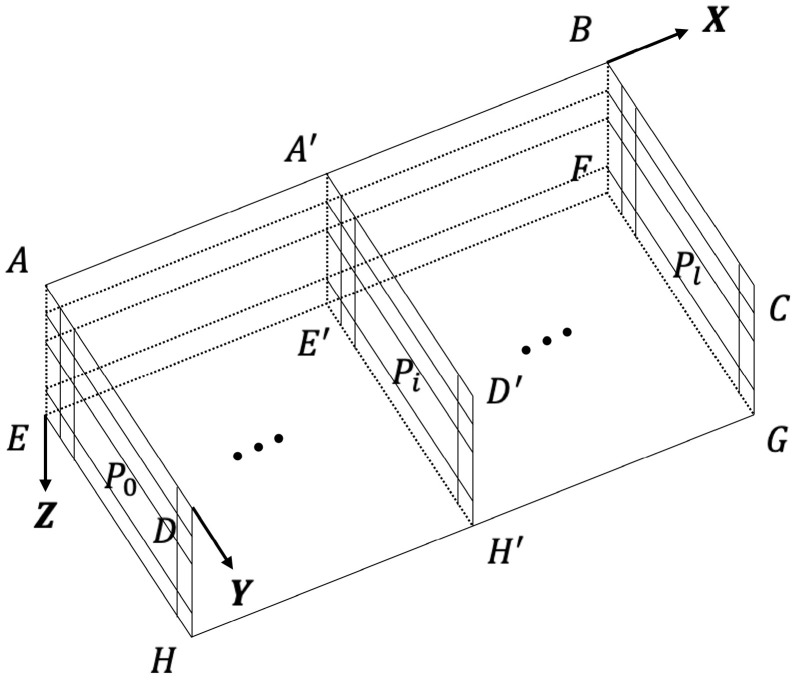
Environment 3D Coordinate System.

**Figure 2 sensors-22-03652-f002:**
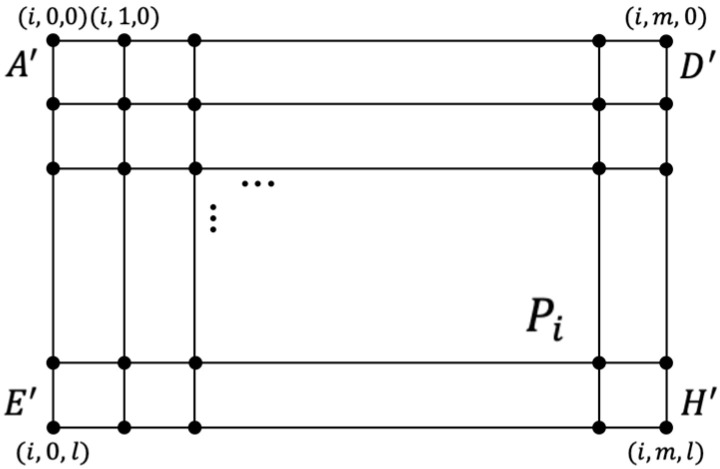
Plan of Coordinate System.

**Figure 3 sensors-22-03652-f003:**
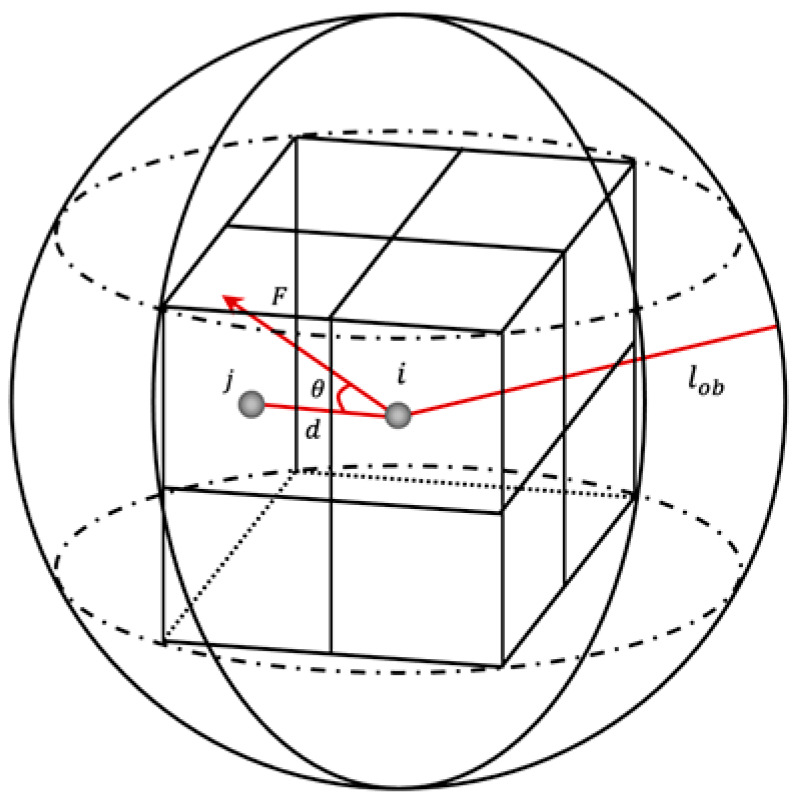
Pheromone diffusion model.

**Figure 4 sensors-22-03652-f004:**
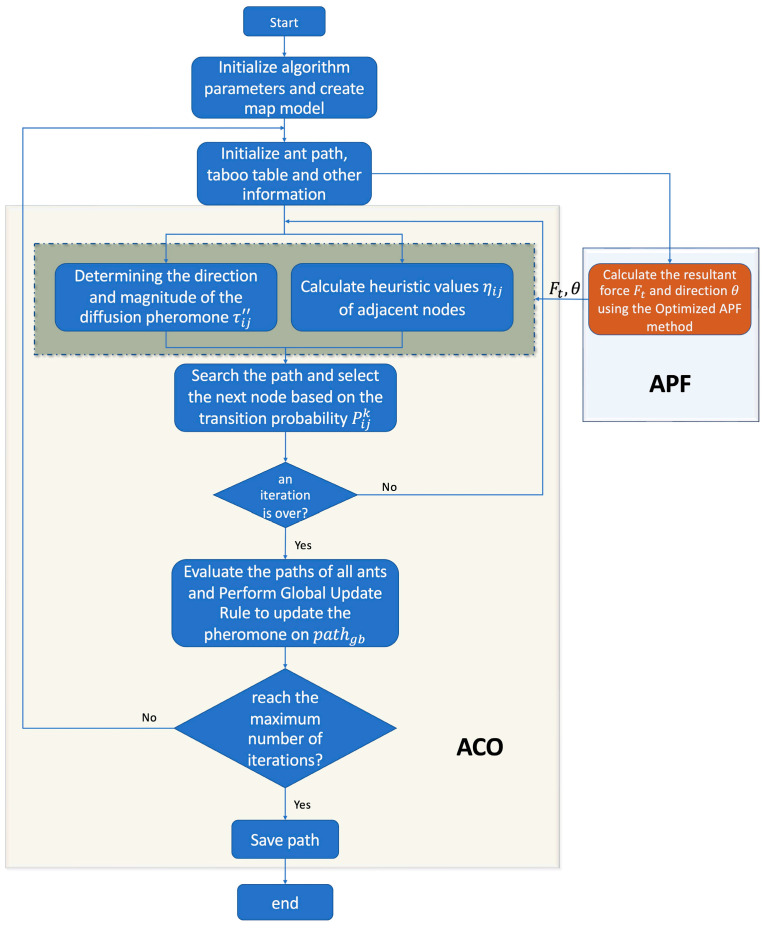
APF-ACO algorithm flow chart.

**Figure 5 sensors-22-03652-f005:**
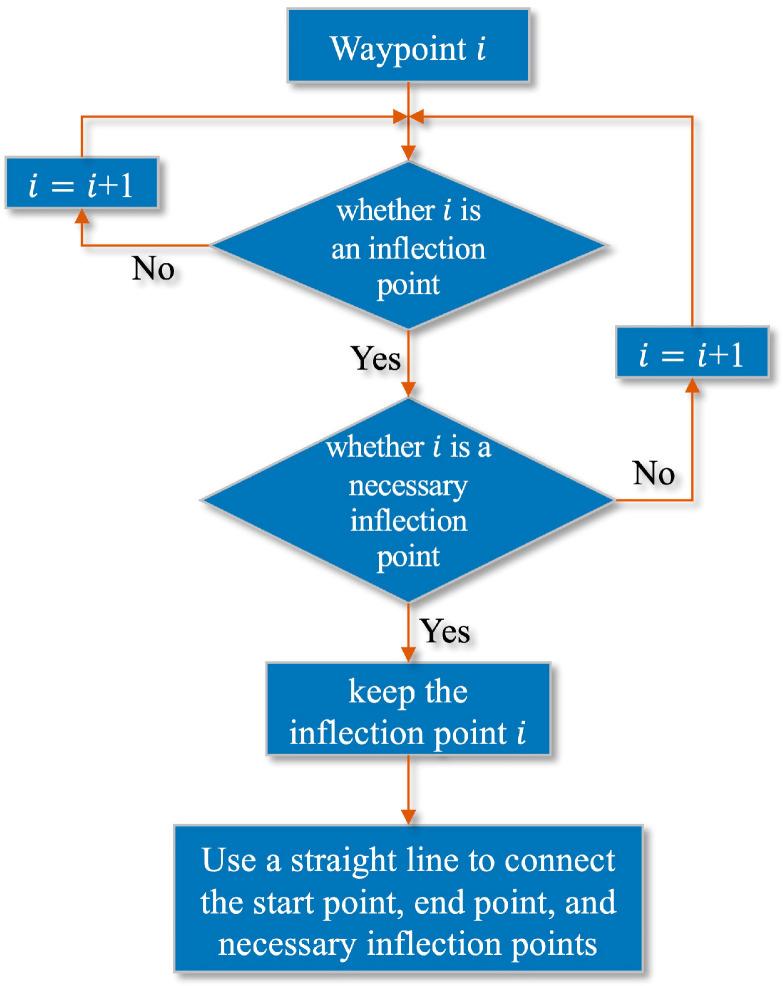
Inflection point optimization algorithm flow chart.

**Figure 6 sensors-22-03652-f006:**
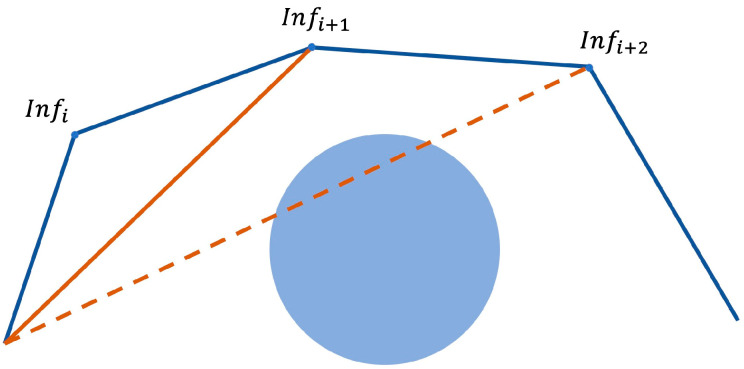
Schematic diagram of the inflection points optimization algorithm.

**Figure 7 sensors-22-03652-f007:**
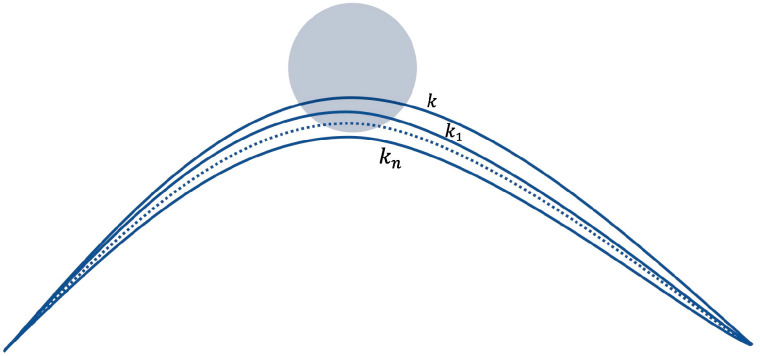
Schematic diagram of the reduced curvature obstacle avoidance algorithm.

**Figure 8 sensors-22-03652-f008:**
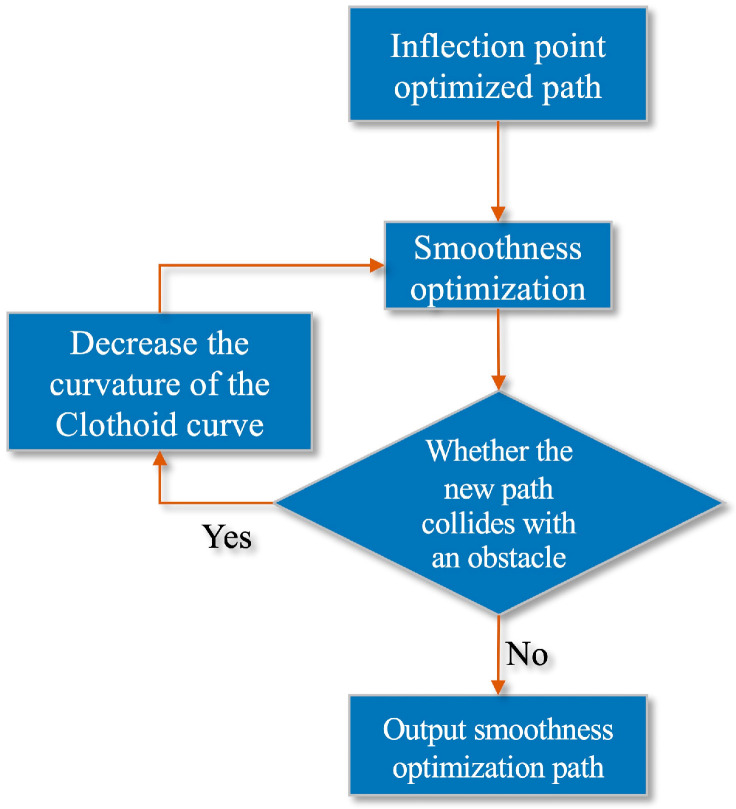
Smooth optimization algorithm flow chart.

**Figure 9 sensors-22-03652-f009:**
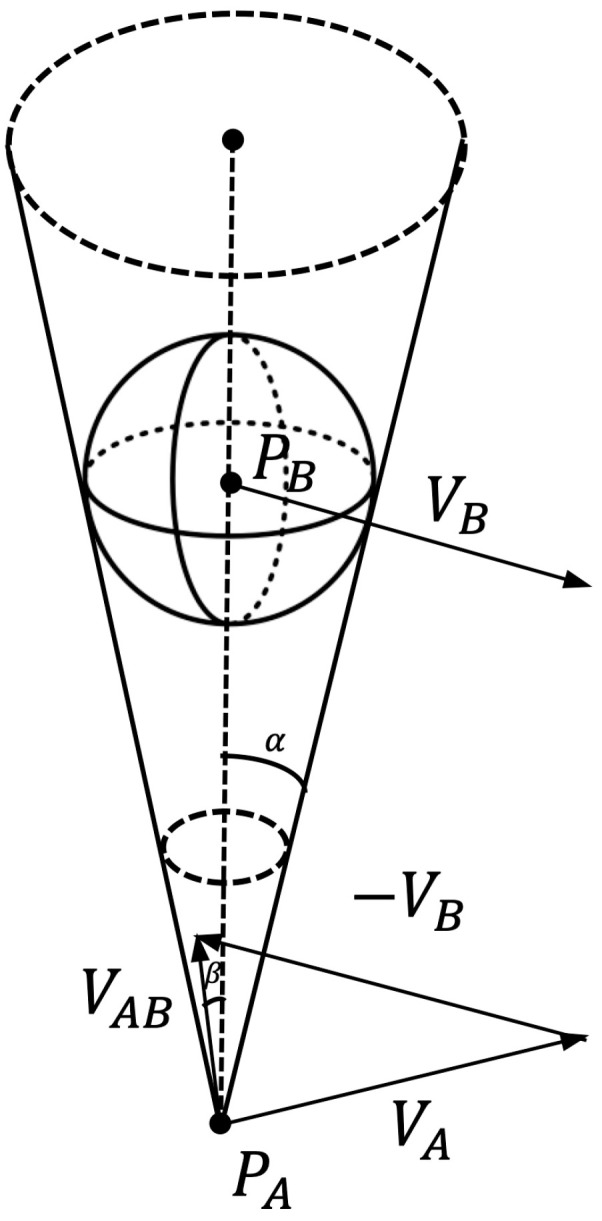
Schematic diagram of 3D dynamic obstacle method.

**Figure 10 sensors-22-03652-f010:**
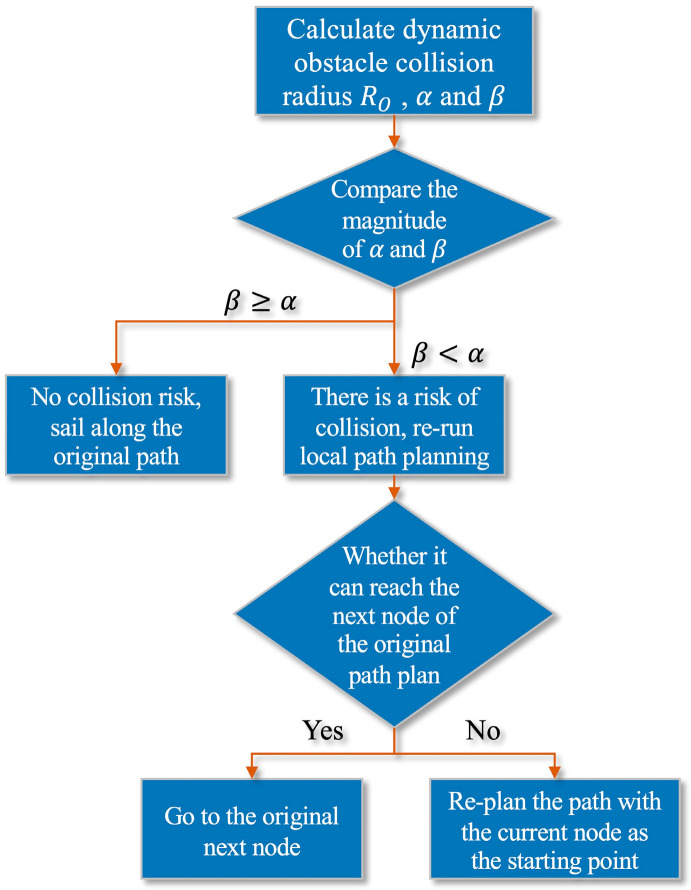
Dynamic obstacle avoidance algorithm flow chart.

**Figure 11 sensors-22-03652-f011:**
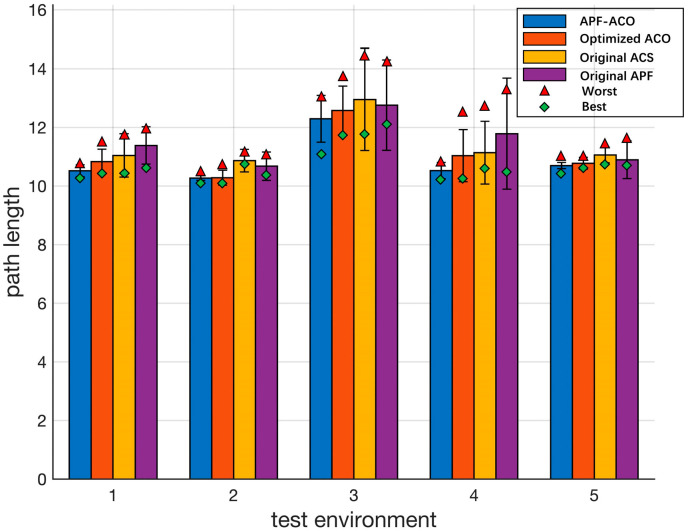
Statistical significance test results.

**Figure 12 sensors-22-03652-f012:**
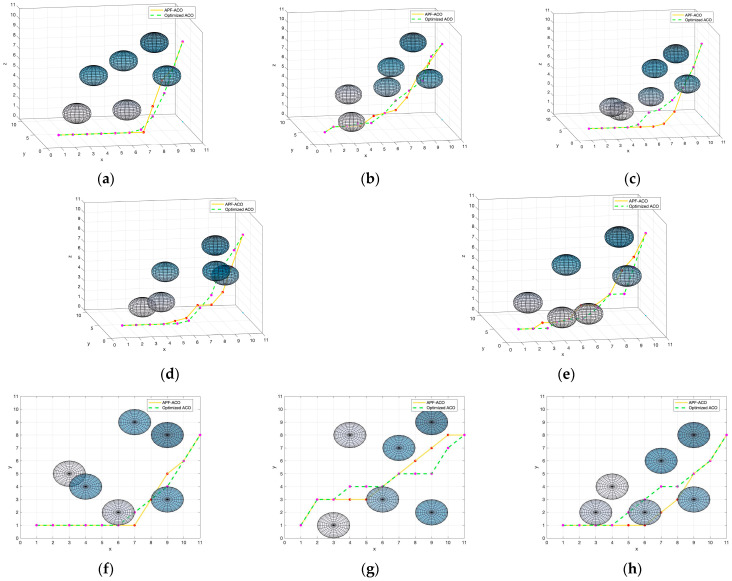
Visualization of path planning results from a certain view (**a**–**e**) and from a top view (**f**–**j**).

**Figure 13 sensors-22-03652-f013:**
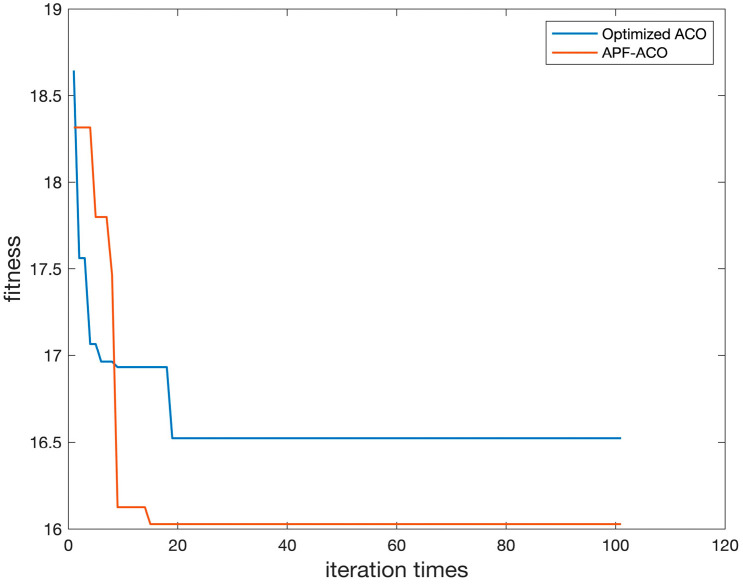
Comparison of Convergence Speed of APF-ACO and Optimized ACO Algorithms.

**Figure 14 sensors-22-03652-f014:**
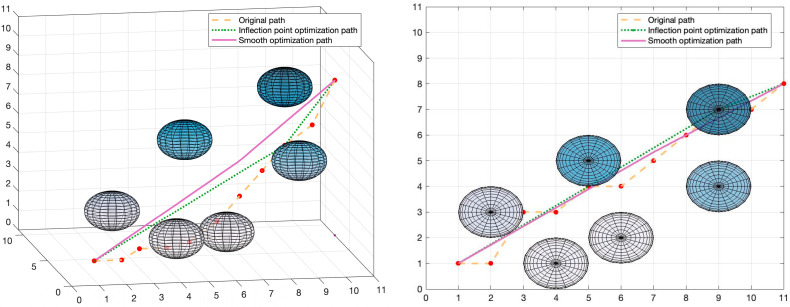
Visualization of path optimization results.

**Figure 15 sensors-22-03652-f015:**
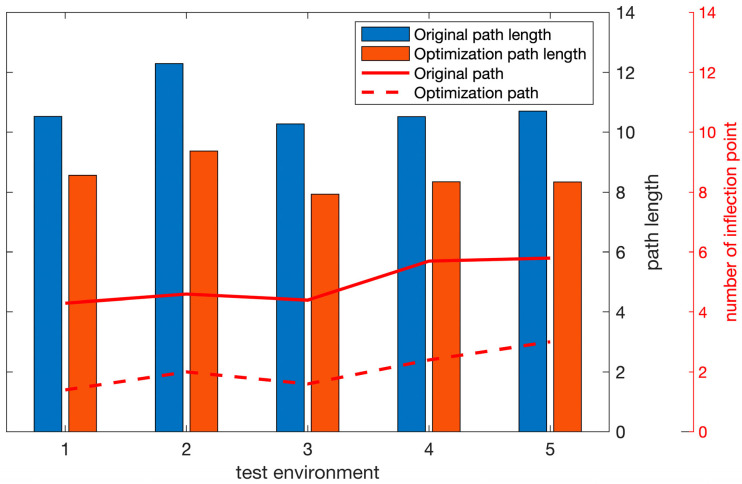
Path Optimization Algorithm Results.

**Figure 16 sensors-22-03652-f016:**
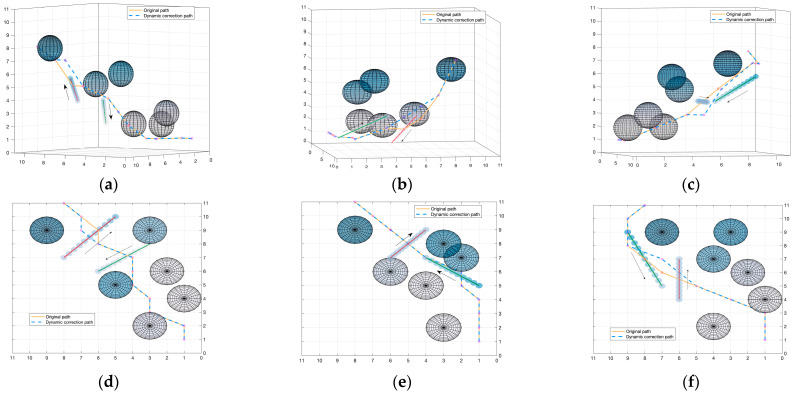
Visualization results of dynamic obstacle avoidance algorithm from a certain perspective (**a**–**c**) and top-down perspective (**d**–**f**).

**Figure 17 sensors-22-03652-f017:**
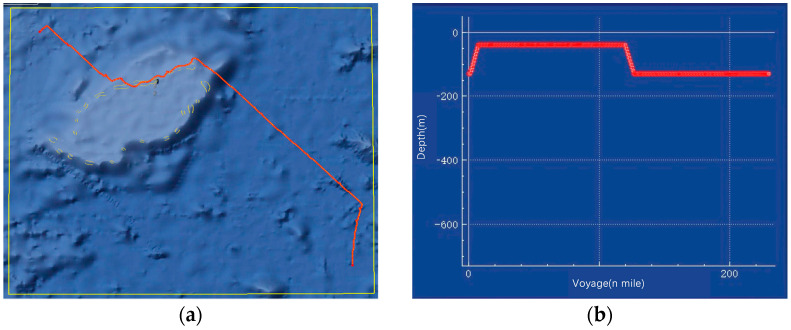
System experimental results, where (**a**) is the overview of the global path, (**b**) is the submarine depth curve, and (**c**) is the path planning from the third person view.

**Table 1 sensors-22-03652-t001:** Algorithm Comparison Summary.

Algorithms	Reference	Advantages	Disadvantages	Vehicle	Improvement
Dijsktra	[1]	global path planning	Marine elements are not considered	Single AUV	first application
$RRT^{*}$	[8]	Adapt to the real environment	slow Convergence	Single AUV	first application
	[26]	Adapt to narrow aisle environments	Not suitable for underwater environments	drone	branch selection rapid exploration random tree
APF	[23]	High robustness	Marine elements are not considered	Single AUV	higher-order sliding mode control
	[25]	Low collision risk, low path vibration	Not suitable for underwater environments	drone	vortex field
PSO	[13]	Fast convergence	Marine elements are not considered	Single AUV	population control
ACO	[24]	Fast convergence and high stability	Marine elements are not considered	Single AUV	alarm pheromone
GWO	[27]	Dynamic obstacle avoidance	2D path planning	Unmanned ships	$D^{*}$ Light algorithm

**Table 2 sensors-22-03652-t002:** The Difference between AUV and Submarine.

Factors	AUV	Submarine
Marine Environmental Elements	ocean current	marine acoustic environment, pycnocline, mesoscale vortex, ocean front, ocean current
Dynamic Characteristics	No need	maximum pitch angle, maximum diving depth, limit turning radius
Size	Usually within 5 m	more than 100 m
Background	civilian mostly	military

**Table 3 sensors-22-03652-t003:** Parameter Settings.

Parameters	Value
APF-ACO	N=100$,\;q_{0}=0.9$, α=β=2.0, ρ=0.1, δ=0.9$,\;k_{p}=1$$,\;k_{rep}=1$
Optimized ACO	N=100, σ=ξ=0.1$,\;q_{0}=0.9$, $\omega_{4}=0.5$, α=β=2.0, ρ=0.1 [26]
The Original ACS	N=100$,\;q_{0}=0.9$, α=β=2.0, ρ=0.1, δ=0.9
The Original APF	$k_{p}=1$ $,\;k_{rep}=1$

**Table 4 sensors-22-03652-t004:** Comparison of the Results Obtained by the Four Algorithms.

Environment	Statistics	The Original APF	The Original ACS	Optimized ACO	APF-ACO
Env1	**Best**	**10.5725**	**10.6732**	**10.2715**	**10.2624**
	Worst	13.2745	12.7655	12.5772	10.8929
	Mean	11.7845	11.1365	11.0360	10.5249
	Std	1.8943	1.0722	0.8898	0.2786
Env2	**Best**	**12.0793**	**11.8988**	**11.8863**	**11.0777**
	Worst	14.0323	14.4365	13.8863	12.8929
	Mean	12.7568	12.9525	12.5772	12.2929
	Std	1.5398	1.7435	0.8312	0.7969
Env3	**Best**	**10.4320**	**10.8561**	**10.1290**	**10.1290**
	Worst	10.8936	11.0753	10.7148	10.3071
	Mean	10.6755	10.8646	10.2818	10.2715
	Std	0.4863	0.3845	0.2540	0.0797
Env4	**Best**	**10.7048**	**10.4853**	**10.4853**	**10.3071**
	Worst	11.8476	11.6849	11.3006	10.7148
	Mean	11.3821	11.0427	10.8335	10.5195
	Std	0.6387	0.7394	0.4204	0.2044
Env5	**Best**	**10.7482**	**10.8472**	**10.7148**	**10.4853**
	Worst	11.4353	11.2785	10.8929	10.8929
	Mean	10.8953	11.0582	10.7742	10.6977
	Std	0.6432	0.2643	0.2044	0.1028

**Table 5 sensors-22-03652-t005:** The Runtime Comparison of the Four Algorithms.

Environment	Runtime (s)			
	The Original APF	The Original ACS	Optimized ACO	APF-ACO
Env1	1.42	1.89	2.16	2.23
Env2	1.27	2.04	2.27	2.19
Env3	1.45	1.78	1.87	1.95
Env4	1.66	1.84	2.04	1.97
Env5	1.39	1.69	1.88	1.92

**Table 6 sensors-22-03652-t006:** Comparison of the number of inflection points between APF-ACO and Optimized ACO.

Algorithms	Number of Inflection Points				
	Test Env1	Test Env2	Test Env3	Test Env4	Test Env5
APF-ACO	4.3	4.6	4.4	5.7	5.8
Optimized ACO	6.5	6.2	5.5	6.2	7.2

**Table 7 sensors-22-03652-t007:** Parameter Settings of the System test.

Algorithms	Value
APF-ACO	N=100$,\;q_{0}=0.9$, α=β= 2.0, ρ=0.1, δ=0.9$,\;k_{p}=1$$,\;k_{rep}=1$

## Data Availability

Not applicable.

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
