# Peer review of "Underwater Submarine Path Planning Based on Artificial Potential Field Ant Colony Algorithm and Velocity Obstacle Method"

_sensors, 2022, doi:10.3390/s22103652_

Round 1

Reviewer 1 Report

The paper combines two types path planning algorithms, the artificial potential filed algorithm and the ant colony algorithm, to achieve the global planning path for submarines. After getting the preliminary path, the paper uses inflection point optimization to reduce the path length and uses Clothoid curve algorithm to smooth the final path. The paper demonstrated that the proposed path planning algorithm had the faster convergence speed and reduced the path length by the virtual simulation and semi-physical simulation. The results show that the method can reduce the path length and smooth the final path. The after-processing ensured the final path without collision.This paper has a good structure and clear study object.

Please find below some comments:

  1. The description of motivation for this paper should be improved. The 3D path planning algorithms have been proposed by many researchers. Why the author think they cannot be used for submarines? Why the author chooses the combination of APF and the ant colony algorithm? What kind of problem is still not be well addressed for 3D path planning?
  2. The path planning algorithm of this paper combined the artificial potential filed algorithm and the ant colony algorithm. The APF may fall in local optimal solution easily. How to deal with this problem in your proposed algorithm? Please give more details.
  3. The computation cost of the proposed algorithm is not clear. It is especially important for local planning. Comparison of computation cost between the proposed algorithm and the other methods should be provided.
  4. In this paper some formulas of parameters is not explained such as formula (20).

Reviewer 2 Report

The paper introduces an approach to path planning for submarines. The text is well-written and interesting, but needs some improvements in order to enhance its clarity and comprehensibility.

Therefore, I recommend to update the manuscript considering the following remarks before the final acceptance:

  1. The phrase “hot spot”, used in line 32 on page 1 and line 72 on page 2 is rather colloquial, it is recommended to replace it with a different phrase.
  2. In line 58 on page 2 “Satisfactory” should not begin with a capital letter.
  3. The phrase “simple easy to implement” in line 62 on page 2 needs rephrasing.
  4. In order to maintain consistency of the whole text, the phrase “et al.” should be written in italics e.g. in lines 66, 74, 77 on page 2.
  5. Comparison of path planning methods mentioned in the introduction on page 2 should be added in a table, considering at least: the type of vehicle, optimization method applied, advantages and disadvantages of a method.
  6. In relation to the sentence “simply using AUV's path planning algorithm for submarine path planning is inappropriate and has hidden dangers” in lines 88-89 on page 2 please show the differences between AUVs and submarines considering the task of path planning in a table for clarity.
  7. Please explain the phrase “path planning results obtained by this algorithm are more advantageous and more stable” on page 3 in line 102, there is a need to add in relation to what other algorithm/algorithms the proposed algorithm is more stable?
  8. Please explain the term “inflection points” (page 3 line 107)
  9. Please consider changing the title of figure 3, as the word “Screenshot” does not seem the most relevant.
  10. On page 5 in line 187 the phrase “heuristic function” should begin with a capital letter.
  11. Please explain the “taboo table” used in the flowchart in Figure 4.
  12. In line 314 on page 8 lack of a space in “XOZin turn”.
  13. Please add the inflection point optimization algorithm and a smooth optimization algorithm flowcharts.
  14. Please add the dynamic obstacle avoidance algorithm flowchart.
  15. On page 11 in line 422 lack of reference “proposed by the reference [ ].”
  16. Please explain how the number of inflection points given in table 2 has been calculated.
  17. Please explain the phrase “knee optimization result” in line 508 on page 15.
  18. Please explain the phrase “composite method” in line 590 on page 17.
  19. Please explain the abbreviation “ACS” used in table 1.

Round 2

Reviewer 1 Report

The authors have addressed the problems well. The manuscript could be considered to be accepted.